# Study of an Ultra-Low-Frequency Inertial Vibration Energy Harvester with a Frequency Up-Conversion Approach

**DOI:** 10.3390/mi16080942

**Published:** 2025-08-16

**Authors:** Jun Chen, Jieliang Xu, Mingjie Guan, Ziqiao Shen, Zilong Cheng

**Affiliations:** 1School of Mechanical and Automotive Engineering, Xiamen University of Technology, Xiamen 361024, China; cj@xmut.edu.cn (J.C.); 2421011037@stu.xmut.edu.cn (J.X.); 2421011008@stu.xmut.edu.cn (Z.C.); 2Pen-Tung Sah Institute of Micro-Nano Science and Technology, Xiamen University, Xiamen 361005, China; 35120231151739@stu.xmu.edu.cn

**Keywords:** piezoelectric energy harvesting, magnetic plucking, frequency up-conversion, flow-induced vibration

## Abstract

For ultra-low-frequency vibration applications, this study focuses on a piezoelectric energy harvesting system with a spring mass system, utilizing magnetic plucking to up-convert the frequency. The proposed spring mass system includes a spring, a magnet mass with a guide rail, and a fixed pulley. The spring mass system responds to external ultra-low-frequency excitation and transfers the vibration to the piezoelectric cantilever beam through the magnets, achieving frequency up-conversion. The theoretical model of the designed piezoelectric energy harvesting system is established, and the effects of magnetic forces and potential energy between the magnets are analyzed. Numerical analysis and experimental studies demonstrate that the proposed piezoelectric energy harvesting system can efficiently achieve frequency up-conversion and generate a higher output power under the conditions of sinusoidal excitation at a frequency of 1 Hz and an amplitude of 40 mm. The system exhibits its highest power output with a magnetic distance of 15 mm, resulting in a maximum output power of 57.35 μW. Finally, to verify the performance of the designed energy harvester in low-velocity water flow, a series of underwater experiments were carried out. The results show that the designed harvester can generate an output power of 23.73 μW with optimal resistance of 250 kΩ at a flow rate of 0.371 m/s. The designed structure is well suited for energy harvesting in flow-induced vibration in low-velocity water flow.

## 1. Introduction

Fluid–structure interaction occurs in slender structures in a transverse flow, possibly causing vortex shedding on the structure’s rear side, which is an important phenomenon in the hydrokinetics field [1]. This vortex shedding may cause structural failure. In early studies, researchers attempted to suppress the vibrations caused by vortex shedding to prevent damage to underwater equipment and structures. In recent years, people have found that this phenomenon can be used for constructive purposes, i.e., to harness hydrokinetic energy, which is plentiful in current flows, such as in river and ocean currents. Systems utilizing horizontal hydrokinetic energy include flow-induced vibration systems and turbine generators. Hydroelectric turbine generators are efficient only when they work in a flow with a velocity higher than 2.5 m/s. However, most ocean current velocities are no larger than 1.5 m/s. Therefore, flow-induced vibration systems are more useful in low-velocity flows. Vortex-induced vibration for an aquatic clean energy (VIVACE) converter was first proposed in 2006. It uses alternating lift to generate vibrations in the cylinders. The vibration energy is converted into electrical energy through an electromagnetic generator and a guide rail. It can harvest hydrokinetic energy from flows as low as 0.343 m/s with substantial efficiency [2,3,4,5,6]. However, when working, the generator and the guide rail should be mounted above the water surface, which is not appropriate in undersea applications.

Another option to harness hydrokinetic energy is to use a piezoelectric transducer. Many studies have been conducted. These studies can be divided into two categories: Firstly, piezoelectric elements are mounted on a beam above water, and the beam is connected with bluff bodies under the water. Secondly, piezoelectric elements and beams are mounted in water and are coated or sealed. For category 1, piezoelectric beams are required to be fixed in some places above the water [7,8,9], which is not possible in wide rivers and oceans. For category 2, piezoelectric elements should be well coated or sealed to be waterproof [10,11,12,13,14], but this is not reliable for long-term use.

In this study, our objective is to design an inertial piezoelectric energy harvester placed inside a bluff body, for example, a cylinder tube. Then, the bluff body is elastically supported in water flow. According to Lv et al. [15], the flow-induced vibration frequency of the cylinder is low, usually in the range 0.5~2 Hz. It is challenging to match the natural frequency of the piezoelectric beam harvester with the low-excitation frequency. Therefore, a frequency up-conversion mechanism is required for the piezoelectric energy harvester inside the tube.

Frequency up-conversion (FUC) mechanisms, which utilize a low-frequency oscillator to absorb energy from external low-frequency excitation and transfer the energy to a high-frequency oscillator, show the highest efficiency and can be used to tackle many challenges [16]. Mechanical plucking and impact are the common methods used to implement frequency up-conversion. A frequency up-conversion mechanism was applied by Cheng et al. [17] based on impact. The end mass of a driving beam periodically impacted the upper and lower piezoelectric cantilever beams. The harvester improved the energy harvesting efficiency in a low-frequency environment. The maximum power output was 0.491 mW.

However, the FUC approaches realized using mechanical plucking and impact are likely to cause fatigue or damage to the structure of the energy harvester and, thus, shorten the service life of the energy harvester. In order to overcome this aforementioned drawback, non-contact magnetic plucking has become a significant focus.

The most significant feature of the non-contact frequency up-conversion mechanism is that it uses a non-contact magnetic force [16]. It can be concluded that the non-contact magnetic force has the following advantages over mechanical impact: (a) It effectively avoids energy loss caused by contact friction and makes less noise. (b) The use of a magnet can tune the working bandwidth of the system. Pillatsch et al. [18] studied effects of a magnetic repulsive force and attractive force as driving forces. According to their results, it was found that using a repulsive force as a driving force resulted in better performance. With a magnetic repulsive force, a bistable oscillator configuration is usually built, which has a double-well restoring force potential. Its periodic interwell oscillation has been recognized as a means to dramatically improve the energy harvesting performance. Numerous studies have probed these subjects with various levels of refinement.

A piezoelectric energy harvester based on a two-stage vibratory structure was proposed by Tang and Li [19]. The first stage picked up the low-frequency vibration of the environment and excited the second stage to vibrate at its resonant frequency. The harvester can provide a substantial improvement in output power in a broad bandwidth. However, the ambient frequency was in the range of 10~30 Hz. Lin et al. [20] proposed a nonlinear magnetic and torsional spring-coupled energy harvester. The proposed energy harvester employed an additional torsion spring to couple the cantilever beam with the base that made the natural frequency of the system easy to adjust. The presence of the 1:3 internal resonance phenomenon in the harvester achieved a frequency enhancement and improved the harvesting performance in low-frequency environments. Their prototype was capable of harvesting the energy in the range of 5–20 Hz. Gao et al. [21] designed a bistable piezoelectric cantilever vibration energy harvester with an elastically supported external magnet. The system could maintain a bistable oscillation state under low-intensity excitation conditions and had better power output performance than the rigidly supported system. Ramezanpour et al. [22] utilized rotating magnets to actuate a piezoelectric bi-morph beam through attractive magnetic interactions. The beams were excited to vibrate at their natural frequencies whenever the magnets passed over the beam. It is possible to increase the generated power by more than ten-times. Mei et al. [23] proposed a piezoelectric energy harvester array for harvesting rotational energy, utilizing magnet-induced nonlinearity and a frequency up-conversion mechanism. The effects of the gap distances, frequency up-conversion and installation configurations on the dynamic characteristics of piezoelectric energy harvesters were investigated. The frequency of the magnetic driving force was in the range of 3.3~10 Hz.

Considering the frequency of flow-induced motion is lower than 2 Hz, the investigated frequency in the above-mentioned studies is still too high. In this study, a spring mass system using magnets as part of the proof mass, with an ultra-low resonant frequency, is needed to resonate with the frequency and offer a magnetic driving force for frequency up-conversion. Another requirement for the mass spring system is that the requisite velocity of the mass should be high enough for interwell oscillation, much higher than those for intrawell or chaotic oscillations.

In view of the flow-induced motion application, the cylinder, the inside piezoelectric cantilever beams, and the mass of the spring mass system all vibrate in the vertical direction. To generate an ultra-low-frequency excitation from the spring mass system, the spring constant should be small and the length of the spring should be large enough. In the radial direction of the cylinder, the diameter of the cylinder would be not large enough. In our design, a pulley is used to solve the problem. The rest of this paper is organized as follows: The system design and mathematical modeling are introduced in Section 2. The theoretical analysis is given in Section 3. In Section 4, the experimental setup and results are shown. The conclusions are drawn in Section 5.

## 2. System Design and Modeling

### 2.1. System Design

To harness the flow-induced motion energy, a tube cylinder is used. The sealed tube forms a waterproof structure to protect the inside energy harvester. A schematic diagram of the energy harvester with the spring mass system is depicted in Figure 1a. The energy harvester system consists of a piezoelectric cantilever beam with a proof mass and a spring mass system. The spring mass system includes a mass, a spring, a guide rail, a slider, and a fixed pulley, which is used to change the spring’s orientation from radial to axial. The cantilever beam is fixed on the side cover, and the piezoelectric elements are surface-bonded near the clamped end. The proof mass consists of a tip mass and a magnet, which are attached to the free end of the cantilever beam. The spring mass system includes another mass and another magnet mounted on the slider, which can move vertically along the guide rail. Since the excitation frequency from the flow-induced motion is lower than 2 Hz, a long spring with a low stiffness is suitable for the spring mass system.

### 2.2. System Model

The equivalent model of the designed system is shown in Figure 1b. The governing equation for an underdamped oscillator excited by the base vibration *x*_0_ (t) is formulated from the physical coordinates, so the dynamic response of the beam can be described using the following governing equations [21]:(1)$$\left\{\begin{array}{l} M_{eq1}{\ddot{x}}_{1}+c_{eq1}{\dot{x}}_{1}+k_{eq1}x_{1}-\Theta V_{1}-F_{1}=-\mu_{1}M_{eq1}{\ddot{x}}_{0}\\ M_{eq2}{\ddot{x}}_{2}+c_{eq2}{\dot{x}}_{2}+k_{eq2}x_{2}+F_{1}=-M_{eq2}{\ddot{x}}_{0}\\ C_{1}^{s}{\dot{V}}_{1}+V_{1}/R_{1}+\Theta {\dot{x}}_{1}=0 \end{array}\right.$$
where *M*_eq1_, *M*_eq2_, *c*_eq1_, *c*_eq2_, *k*_eq1_ and *k*_eq2_ represent the equivalent mass, equivalent damping and equivalent stiffness of the two subsystems, respectively; *x*_1_ and *x*_2_ are the relative displacements of proof mass A and proof mass B, respectively; *V*_1_ is the output voltage of the piezoelectric cantilever beam; *μ*_1_ is the correction factor of the forcing function; Θ is the piezoelectric coupling coefficient; *R*_1_ is the load resistance. *M*_eq1_, *M*_eq2_, *c*_eq1_, *c*_eq2_, *k*_eq1_, and Θ are given by Equations (2)–(7), respectively:(2)$$M_{eq1}=m_{A}+33m_{beam}/140$$(3)$$M_{eq2}=m_{B}+m_{slider}$$(4)$$k_{eq1}=6E_{p}I/L^{2}\left(2L+1.5l_{B}\right)$$(5)$$I=2\left(w_{b}t_{p}^{3}/12+w_{b}t_{p}{\left(t_{p}+t_{b}\right)}^{2}/4\right)+E_{b}w_{b}t_{b}/12E_{p}$$(6)$$c_{eq}=2M_{eq1}\xi_{r}\omega_{r}$$(7)$$\Theta =e_{31}\psi_{r}^{\prime }(t_{b}+t_{p})/2$$
where *m*_A_, *m*_B_, *m*_beam_ and *m*_slide_ represent mass of the proof mass A, proof mass B, piezoelectric cantilever beam and slider; *I* represents the rotational inertia; *E*_b_ represents the elasticity modulus of cantilever substrate; *E*_e_ represents the elasticity modulus of piezoelectric element; *ξ*_r_ represents the mechanical damping ratio; *ω*_r_ represents the structural natural frequency of the piezoelectric cantilever beam; *e*_31_ represents the piezoelectric constant; and ψ_r_’ represents the spatial derivative of the mechanical mode shape. Other parameters are shown in Table 1.

### 2.3. Magnetic Force and Magnetic Potential Energy

In the designed structure, the effective surface of the two magnets varies with their relative positions. As a result, utilizing a static model to analyze the magnetic force will be inaccurate. To address this, an expression for the magnetic force is derived through vector differentiation. It is assumed that Magnets A and B can be simplified as magnetic dipole A and B, respectively. Firstly, the magnetic force ***F***_21_ exerted by the magnetic dipole B of the spring mass system on the magnetic dipole A at the tip of the piezoelectric cantilever beam is derived. The geometric analytical model of the two magnets is shown in Figure 2. In Figure 2, ***m*_1_** and ***m*_2_** are the magnetic moment vectors for magnetic dipole A and B, ***r*** is the vector from the center of the Magnet A to the center of the Magnet B, and *d*_1_ is the horizontal distance between the two magnetic dipoles. Point A’ is the projection point of Magnet A on the horizontal extension line of the cantilever beam. Point B’ is the intersection of the cantilever beam’s horizontal extension line with *r*. *d* is the distance between A’ and B’. In addition, α, β, and θ represent the angles between ***m*_1_** and ***r***, ***m*_2_** and ***r***, and ***m*_1_** and the horizontal line, respectively.

The magnetic flux density ***B***_21_ generated by Magnet B at Magnet A can be expressed as:(8)$$B_{21}=\frac{\mu_{0}}{4\pi r^{3}}\left(3\left(m_{2}\cdot \hat{r}\right)\hat{r}-m_{2}\right)$$
where ***r*** = −*d*_1_ ***i*** + (x_1_ − x_2_) ***j***, *r* is the length of ***r***, and $\hat{r}$ is the direction vector of ***r***.

Then, the interaction force of Magnet A on Magnet B is:(9)$$F_{21}=\nabla \left(m_{1}\cdot B_{21}\right)$$
where ∇ is the vector gradient operator, $m_{1}=m_{1}\cos\theta i+m_{1}\sin\theta j$, and $m_{2}=-m_{2}j$. By substituting the gradient functions $\nabla \frac{1}{r^{n}}=-\frac{nr}{r^{n+2}}$,$\nabla \left(v_{1}\cdot r\right)=v_{1}$ and $\nabla \left(ab\right)=a\left(\nabla b\right)+b\left(\nabla a\right)$ [24] into Equation (9), where *a* and *b* represent scalar function, the expression of the magnetic force can be obtained:(10)$$\begin{array}{cc} F_{21} & =\nabla \left(m_{1}\cdot \frac{\mu_{0}}{4\pi r^{3}}\left(3\left(m_{2}\cdot \hat{r}\right)\hat{r}-m_{2}\right)\right)\\ & =\frac{\mu_{0}}{4\pi }\nabla \left(\frac{3\left(m_{2}\cdot r\right)\left(m_{1}\cdot r\right)}{r^{5}}-\frac{m_{1}\cdot m_{2}}{r^{3}}\right)\\ & =\frac{3\mu_{0}}{4\pi r^{5}}\left(\left(m_{2}\cdot r\right)m_{1}+\left(m_{1}\cdot r\right)m_{3}+\left(m_{1}\cdot m_{2}\right)r-\frac{5\left(m_{2}\cdot r\right)\left(m_{1}\cdot r\right)r}{r^{2}}\right) \end{array}$$

Since Magnet B and the piezoelectric cantilever beam only vibrate in the vertical direction, only the relationship in the vertical direction is considered in this study. Then, the magnetic force of Magnet A on Magnet B in the vertical direction is:(11)$$F_{21v}=\frac{3\mu_{0}m_{1}m_{2}}{4\pi r^{4}}\left(\cos\beta \sin\theta -\cos\theta \sin\beta -5\cos\beta \cos\alpha \sin\beta \right)$$

The following trigonometric relations can be deduced from Figure 2:(12)$$\left\{\begin{array}{l} \sin\theta =\frac{x_{1}}{\sqrt{L^{2}+{x_{1}}^{2}}},\cos\theta =\frac{L}{\sqrt{L^{2}+{x_{1}}^{2}}}\\ \sin\beta =\frac{x_{1}-x_{2}}{\sqrt{{d_{1}}^{2}+{\left(x_{1}-x_{2}\right)}^{2}}},\cos\beta =\frac{d_{1}}{\sqrt{{d_{1}}^{2}+{\left(x_{1}-x_{2}\right)}^{2}}}\\ \cos\alpha =\cos\left(180^{\circ }-\left(\theta +\beta \right)\right)=\sin\theta \sin\beta -\cos\theta \cos\beta \end{array}\right.$$

The potential energy exerted by Magnet B on Magnet A can be defined as $U_{m}=-B_{21}\cdot m_{1}$, and the derivation can be simplified based on Equation (8). The process is as follows:(13)$$\begin{array}{cc} U_{m} & =-B_{21}\cdot m_{1}\\ & =-\frac{\mu_{0}}{4\pi r^{3}}\left(3\left(m_{2}\cdot \hat{r}\right)\hat{r}-m_{2}\right)\cdot m_{1}\\ & =\frac{\mu_{0}m_{1}m_{2}}{4\pi r^{3}}\left(-\cos\theta -3\cos\beta \cos\alpha \right)\\ & =\frac{\mu_{0}m_{1}m_{2}}{4\pi r^{3}}\left(-\frac{L}{\sqrt{L^{2}+{x_{1}}^{2}}}-\frac{3d_{1}}{\sqrt{{d_{1}}^{2}+{\left(x_{1}-x_{2}\right)}^{2}}}\cdot \frac{x_{1}\left(x_{1}-x_{2}\right)-Ld_{1}}{\sqrt{L^{2}+{x_{1}}^{2}}\sqrt{{d_{1}}^{2}+{\left(x_{1}-x_{2}\right)}^{2}}}\right)\\ & =\frac{\mu_{0}m_{1}m_{2}}{4\pi }\left(\frac{-\left(L+3d_{1}\right)x_{1}^{2}-Lx_{2}^{2}+\left(3d_{1}+2L\right)x_{1}x_{2}+2d_{1}^{2}L}{{\left(\sqrt{d_{1}^{2}+{\left(x_{1}-x_{2}\right)}^{2}}\right)}^{5}\sqrt{L^{2}+x_{1}^{2}}}\right) \end{array}$$

## 3. Theoretical Analysis

### 3.1. Potential Energy Analysis

The designed energy harvesting system consists of a piezoelectric cantilever beam and a magnetic plucking system. The total potential energy of the piezoelectric cantilever beam *U* includes its elastic potential energy and magnetic potential energy *U_m_*, which can be written as:(14)$$U=\frac{1}{2}k_{eq1}x_{1}^{2}+U_{m}$$

By substituting the system parameters in Table 1 into Equation (14), the total potential energy varies with the displacement *x*_2_ of Magnet B. By setting the displacement *x*_2_ as −30 mm, −10 mm, −3 mm, 0 mm, 3 mm, 10 mm and 30 mm, the potential energy curves of the piezoelectric cantilever beam are shown in Figure 3. It is shown that the potential energy curves vary with the displacement *x*_2_. When *x*_2_ = 0, the potential energy barrier is 0.06 J, and the system can be regarded as a magnetic plucking energy harvester with the magnet at the equilibrium position. The piezoelectric cantilever beam needs to cross through the potential barrier to generate interwell oscillation [25]. When the external excitation frequency is as low as 1 Hz and the maximum acceleration is lower than 0.3 g, the magnetic plucking energy is less than the potential energy barrier, and the piezoelectric cantilever beam will exhibit an intrawell oscillation and stay either in the upper- or lower-potential well. As a result, the output power is low during intrawell oscillation. It is shown that when the displacement *x*_2_ is located in the lower-potential-energy well, the potential energy curve goes downward from the high energy bound to the lower-potential-energy well, which makes it easier to cross through the shallow-potential-energy barrier to the upper-potential-energy well. Conversely, when the displacement *x*_2_ is located in the upper-potential-energy well, the potential curve is more likely to cross through the shallow-potential-energy barrier from the upper-potential-energy well to the lower-potential-energy well. Under this condition, the proposed harvester may exhibit periodic interwell oscillations.

For a given value of distance *d*_1_, there is *x*_1_/*x*_2_ = *d*/(*d*_1_ − *d*). Any value of d between 0 and *d*_1_ will correspond to a set of displacements *x*_1_ and *x*_2_. Therefore, different value of *d* can represent variation in *x*_1_ and *x*_2_. The potential energy associated with each set of *x*_1_ and *x*_2_ forms a corresponding potential energy curve. Therefore, it is essential to analyze the effect of parameter *d* on the potential energy. This approach provides a more intuitive analysis of how the potential energy varies with magnet distance *d*_1_. By setting *d/d*_1_ as 1, 0.75, 0.5, 0.25, and 0.01, the corresponding potential energy curves are derived from the potential energy function, as illustrated in Figure 4. It can be seen that the potential energy barrier for the beam reaches a maximum of 0.173 J and a minimum of 0.06 J. In contrast, the spring system displays a different energy profile.

It can be seen that the potential barrier of the spring mass system decreases, while that of the piezoelectric cantilever beam increases with increasing *d*. The potential energy curves of the spring mass system and the piezoelectric cantilever beam corresponding to a certain *d* exhibit a lower potential barrier, making it easier to achieve bistable oscillation, which may result in stronger interwell oscillation. The spring mass system can also cross through the potential energy barrier via magnetic plucking in this situation.

When the magnetic distance is set to *d*_1_, the potential well of the spring mass system is the deepest, and the system is equivalent to an energy harvester with a fixed magnet. Assuming there is an initial magnetic distance that the harvester can cross through the highest-potential barrier, other situations with shallower barriers will enable bistable transition oscillations.

When *d* = *d*_1_, the potential energy curves corresponding to different initial magnetic distance *d*_1_ are shown in Figure 5. When the magnetic distance *d*_1_ = 7 mm, the potential energy barrier is 0.09 J. It gradually decreases with the increasing magnetic distance until the system transitions to the monostable state. The results indicate that negative linear stiffness occurs exactly at an initial magnetic distance of 15 mm, and potential energy starts to exhibit two stable equilibrium points, separated by an unstable equilibrium point. This critical magnetic distance is the magnetic distance for optimal system performance [26]. At the critical magnetic distance of 15 mm, the spring mass system forms interwell oscillations under ultra-low-frequency excitation. Energy is transferred to the piezoelectric cantilever beam via magnetic plucking, enabling oscillations in the piezoelectric cantilever beam as well. Bistable transitions will increase the output power and electromechanical conversion efficiency of the system [24].

### 3.2. Dynamic Response

In this study, ultra-low-frequency ambient excitation is simulated using a sine function with an amplitude of 40 mm and a frequency of 1 Hz. The dynamic analysis of the established system model is executed by employing the Runge–Kutta algorithm, and the ode45 solver of MATLAB R2024a is used to observe the dynamic response of *x*_1_(t), *x*_2_(t), and *V*(t). Firstly, the output voltage responses at different magnetic distances of 14 mm, 15 mm, 17 mm and 25 mm were investigated, and their output voltages are shown in Figure 6.

It can be observed that the optimal output response of this energy harvesting system is achieved at a magnetic distance of 15 mm, where the average value of the peak-to-peak voltage is the highest. The theoretical maximum output power of the system is calculated to be 59.81 μW by the equation $P=U^{2}/4R_{1}$. Figure 7 illustrates the dynamic response of the system under external excitation conditions of *f* = 1 Hz, A = 40 mm, and *d*_1_ = 15 mm. Figure 7a displays the displacement responses of the tip mass at the end of the beam *x*_1_ and the mass in the spring mass system *x*_2_, as well as the output voltage response of the piezoelectric element. Figure 7b shows the phase portraits of displacement *x*_1_, while Figure 7c illustrates its amplitude–frequency spectrum. It can be seen that the system is undergoing multiple periodic motion, and it has a maximum amplitude at a frequency of 6 Hz under an excitation frequency of 1 Hz. Furthermore, the presence of the 1:6 internal resonance phenomenon in the harvester considerably enhances the frequency response, leading to significant improvements in energy harvesting under low-frequency ambient excitation.

## 4. Experimental Setup and Results

### 4.1. Experimental Setup

Figure 8 shows an experimental schematic diagram and the prototype of the proposed harvester. In Figure 8a, the microcontroller produces signals to the optocoupler. After amplification and isolation by the optocoupler module, the signals are sent to the servo driver modeled DO-1000C/50A (MIGE Motor Co., Ltd., Hangzhou, China) to drive the motor. Consequently, the lead screw is driven to rotate, causing the sliding block and the shaker plate to move up and down. The prototypes of the harvester and its spring mass system are depicted in Figure 8b,c, respectively. On the left side, an aluminum base beam is clamped at the base, and a piezoelectric patch modeled DH-P5 (Xinchang Dihui Electronics Co., Ltd., Xinchang, China) is bonded close to the clamping end. Mass A and Magnet A are attached at the end of the beam. Mass B and Magnet B are fixed on the slider, enabling it to slide vertically along the guide rail. A rigid rope passes through the fixed pulley, and the two ends are, respectively, connected to Mass B and the horizontal spring. The spring has a size of 0.3 mm × 3 mm × 80 mm, and the spring stiffness can be calculated using $k={G_{s}d}_{s}^{4}/(8D_{m}^{3}N_{c})$ where *G_s_* is the shear modulus, *d_s_* is the spring wire diameter, *D_m_* is the center diameter, and *N_c_* is the number of coils. During the experiment, an oscilloscope modeled DS1104Z (Rigol Co., Ltd., Beijing, China) was utilized to measure the open-circuit output voltage of the piezoelectric patch. The motor is model 110ST-M06030 (MIGE Motor Co., Ltd., Hangzhou, China), and the microcontroller is model STM32F106.

### 4.2. Experimental Results

In order to experimentally verify the critical magnetic distance in the system, the performance of the energy harvester is investigated at magnetic distances *d*_1_ ranging from 10 mm to 32 mm. The external sinusoidal excitation is set at a frequency of 1 Hz. By adjusting the magnetic distance *d*_1_, four typical output voltages are presented in Figure 9a. The output voltage was analyzed to establish the correlation between the output power and the magnetic distance at various excitation amplitudes, as depicted in Figure 9b. The output voltage and output power of the system increase at first and then decrease after the peak as the magnetic distance increases. At an excitation amplitude of 40 mm, the output power at different magnetic distances is significantly higher than that at other amplitudes. Furthermore, with a decrease in the excitation amplitude, the magnetic distance corresponding to the maximum output power gradually increases. Therefore, when weakening the external excitation level, it is necessary to increase the magnet distance to achieve the magnetic plucking effect.

The output powers of the energy harvester with different magnetic distances were analyzed under an excitation amplitude of 40 mm. When the magnetic distance is 10 mm, 12 mm, and 14 mm, the excessive magnetic plucking prevents the spring mass system from crossing through the potential barrier. Under this condition, the piezoelectric cantilever beam may exhibit low-energy intrawell vibrations or occasional interwell oscillations, resulting in small output voltages. The maximum output power is 5.23 μW, 4.93 μW and 16.96 μW, respectively. At a magnetic distance of 15 mm, the structure demonstrates optimal performance. The spring mass system resonates with external excitation, and the piezoelectric cantilever exhibits periodic intrawell oscillations under the influence of the magnetic force. The peak-to-peak value of the output voltage reaches 36 V, with a maximum output power of 57.35 μW. It is consistent with the theoretical analysis results presented in the previous section. When *d*_1_ exceeds 15 mm, the magnetic force weakens with an increase in the magnetic distance, leading to a noticeable decrease in the output voltage and output power.

In Figure 10, a comparison is made between the experimental and theoretical output voltages under sinusoidal excitation with an amplitude of 40 mm and a frequency of 1 Hz, considering the optimal initial magnetic distance of 15 mm. The output voltage frequencies are both at 6 Hz, and the voltage amplitudes are very close, resulting in good waveform alignment. However, the theoretical output voltage is slightly higher than the actual value, which may be due to the omission of frictional forces from mechanical components such as the guide rail, slider, and fixed pulley in the theoretical analysis process.

Figure 11 depicts a comparison of the output voltages with and without magnetic plucking. It is evident that the peak-to-peak output voltage increases by 3.6-times, and the frequency of the piezoelectric beam increases by 6-times. When the magnetic distance is 15 mm, the spring mass system and piezoelectric cantilever beam can vibrate, respectively, at frequencies of 1 Hz and 6 Hz. The results show that the designed system generates 1:6 internal resonance. The output power of the system is significantly improved from 0.88 μW to 57.35 μW with magnetic plucking. 

A comparative sinusoidal excitation experiment was conducted under different excitation frequencies (0.8 Hz, 1 Hz, and 1.2 Hz) and four excitation amplitudes (25 mm, 30 mm, 35 mm, and 40 mm) at the critical magnetic distance of 15 mm. As shown in Figure 12a, it is clearly shown that the output power increases with an increment in excitation amplitude at different excitation frequencies. When the excitation amplitude is less than 35 mm, the higher the excitation frequency, the higher the output power. However, when the excitation amplitude reaches 40 mm, the output power of the excitation amplitude of 1 Hz begins to exceed the output power of the excitation frequency of 1.2 Hz and is 1.9-times higher than the latter. When the excitation amplitude is 40 mm, the output power is plotted against the excitation frequency (0.8 Hz, 1 Hz, 1.2 Hz and 1.5 Hz), as shown in Figure 12b. It was observed that at a frequency of 0.8 Hz, the spring mass system and the piezoelectric cantilever beam are unable to overcome the potential barriers. As a result, vibrations are at a low level, and the output power is only 1.23 μW. However, at excitation frequencies of 1 Hz, 1.2 Hz and 1.5 Hz, the output power reaches 57.35 μW, 38.81 μW and 58.06 μW, respectively. These external excitations can provide enough energy so that the spring mass system and the piezoelectric cantilever beam can successfully cross through the potential barriers to realize periodic interwell oscillations.

Figure 12b,c show the output voltage curves with and without magnetic plucking under sinusoidal excitation at a frequency of 1 Hz corresponding to four different excitation amplitudes: 25 mm, 30 mm, 35 mm, and 40 mm (equivalent acceleration of 0.1 g, 0.12 g, 0.14 g, and 0.16 g, respectively). The output voltage curves show a gradual increase with the increasing excitation amplitude. Without magnetic plucking, the output voltages are low due to the fact that the external excitation frequency is significantly lower than the natural frequency of the piezoelectric cantilever beam, resulting in only a weak vibration of the cantilever beam. When magnetic plucking is introduced, the output voltages for different amplitudes are greater than those without magnetic excitation. After the excitation amplitude reaches 35 mm (0.14 g), the system exhibits interwell oscillations, which significantly increase the peak-to-peak output voltage and output power. 

In order to verify the performance of the designed piezoelectric energy harvester in water flow, the proposed harvester was placed inside a cylinder tube and two end covers were used to seal the cylinder tube, as shown in Figure 13b. The cylinder tube has an outer diameter of 100 mm, an inner diameter of 90 mm, and a length of 800 mm, which is made of PMMA, allowing for visualization of the inside parts in the tube. The cylinder tube is elastically supported by springs, as shown in Figure 13b. The sealed piezoelectric energy harvester is placed in the water channel, and the water flow made the tube move elliptically in the XOZ plane; the designed piezoelectric energy harvester can effectively capture the vertical vibration energy.

The output voltages of the piezoelectric energy harvester with and without magnetic plucking at different flow rates were obtained, as shown in Figure 14. From Figure 14, it can be seen that the output voltages of the harvester with magnetic plucking were greater than those without magnetic plucking. As the flow rate increased from 0.313 m/s to 0.431 m/s, the vibration acceleration generated from flow-induced vibration in the low-velocity water flow ranged from 0.024 g to 0.126 g. The peak-to-peak voltage of the energy harvester with the spring mass system increased from 14.2 V to 29.6 V and then decreased to 3.4 V. When the flow rate exceeds 0.371 m/s, the designed spring mass system is able to respond to the ultra-low-frequency flow-induced vibration from water flow and excites the piezoelectric beam to vibrate at its resonant frequency with magnetic plucking. Additionally, the system can stably realize bistable characteristics, leading to a significant enhancement in output power. Furthermore, different values of load resistance from 100 kΩ to 400 kΩ were connected with the piezoelectric element. Figure 14f indicates that the harvester achieved a maximum power of 23.73 μW at an optimal resistance of 250 kΩ. In summary, the proposed piezoelectric energy harvesting system with a spring mass system exhibits good performance in capturing energy from flow-induced vibration in the low-velocity water flow.

## 5. Conclusions

The vibration frequencies of the bluff body in the flow-induced vibration in low-velocity flow are usually under 2 Hz, and the space of the bluff body is limited. To effectively harvest the vibration energy in ultra-low frequency, a piezoelectric energy harvesting system with a spring mass system is presented in this study. The frequency up-conversion mechanism is applied in the system, converting low-frequency vibrations into high-frequency oscillation and enabling reciprocating interwell oscillation.

In this study, a theoretical model of the energy harvester with the spring mass system is developed using Newton’s second law, and the magnetic force and magnetic potential energy between two magnets are derived. The bistable characteristics of the system are numerically analyzed. The frequency up-conversion performance of the energy harvester is investigated through a series of numerical simulations and experiments. The experimental results show that, under a sinusoidal excitation with a frequency of 1 Hz and an amplitude of 40 mm, the system achieves its maximum output power of 57.35 μW at a magnetic distance of 15 mm, about 68-times higher than the system without magnetic plucking. At this distance, the system is capable of achieving an internal resonance of 1:6, and the output voltage frequency of the system is about six-times of the excitation frequency, which shows a notable frequency up-conversion effect. In addition, the effect of excitation on the output power of the system is also verified over a range of excitation frequencies from 0.8 Hz to 1.5 Hz and excitation amplitudes from 25 mm to 40 mm, respectively. The results show that the spring mass system and the piezoelectric cantilever beam can successfully cross the potential barriers and realize periodic interwell oscillations under an excitation amplitude of 40 mm and an excitation frequency in the range of 1 Hz to 1.5 Hz. Finally, it is verified that the designed system can generate an output power of 23.73 μW with an optimal resistance of 250 kΩ at a flow rate of 0.371 m/s.

## Figures and Tables

**Figure 1 micromachines-16-00942-f001:**
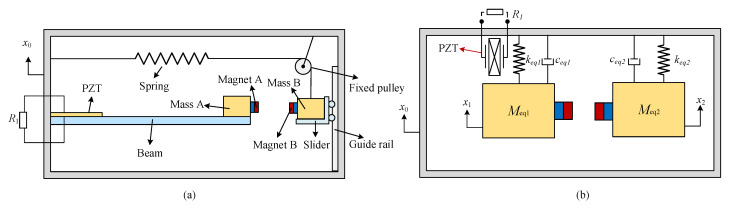
The proposed energy harvester: (**a**) schematic diagram; (**b**) equivalent model.

**Figure 2 micromachines-16-00942-f002:**
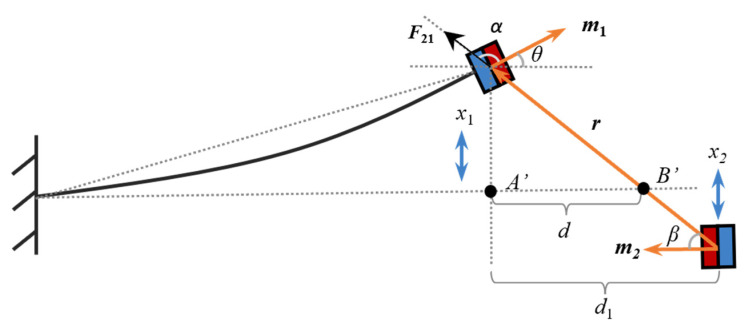
Geometric analytical model of the two magnets.

**Figure 3 micromachines-16-00942-f003:**
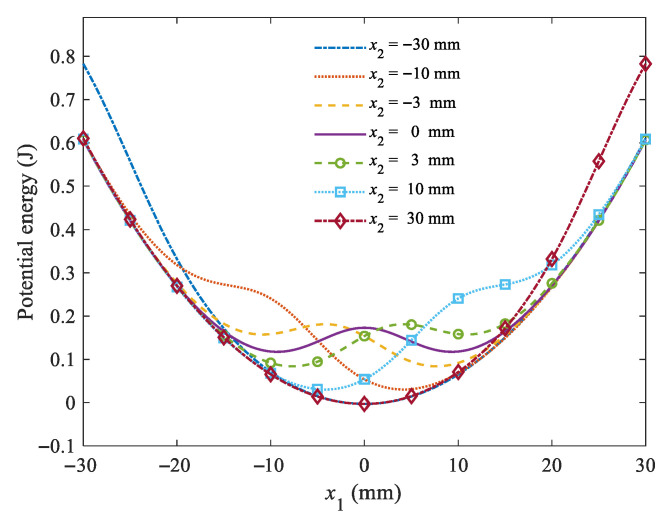
Potential energy curves of the piezoelectric cantilever beam under different displacements.

**Figure 4 micromachines-16-00942-f004:**
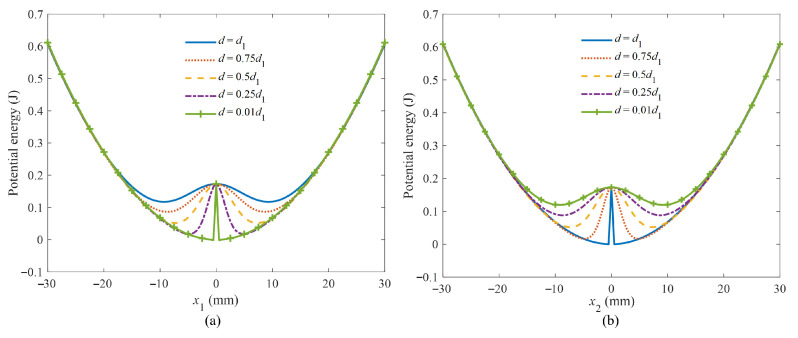
Potential energy of two subsystems: (**a**) piezoelectric cantilever beam; (**b**) spring mass system.

**Figure 5 micromachines-16-00942-f005:**
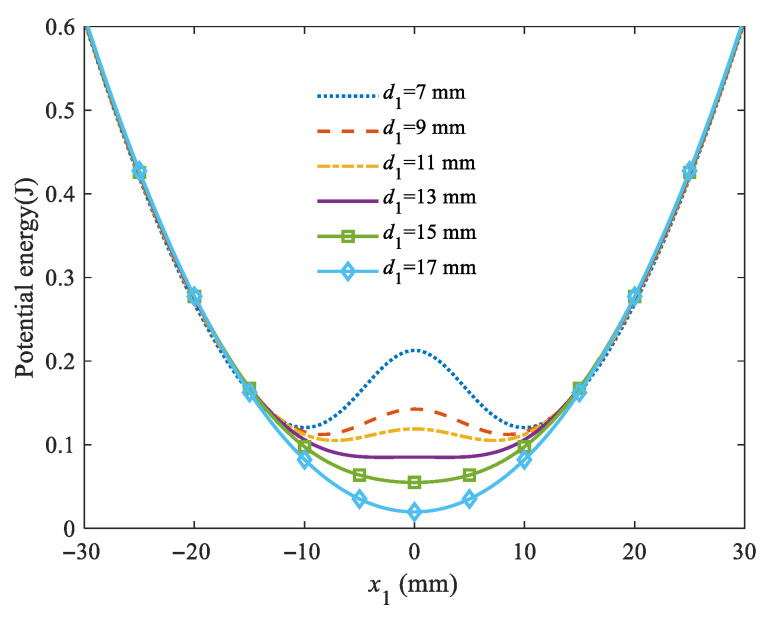
Potential energy curves corresponding to different initial magnetic distances *d*_1_.

**Figure 6 micromachines-16-00942-f006:**
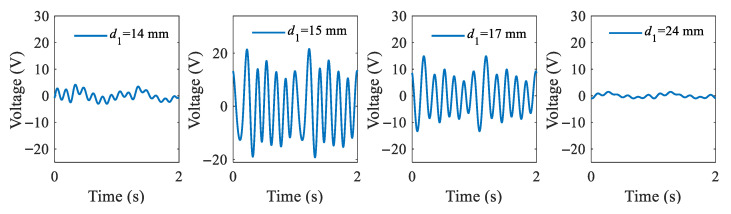
Output voltage waveforms of the system at different magnetic distances.

**Figure 7 micromachines-16-00942-f007:**
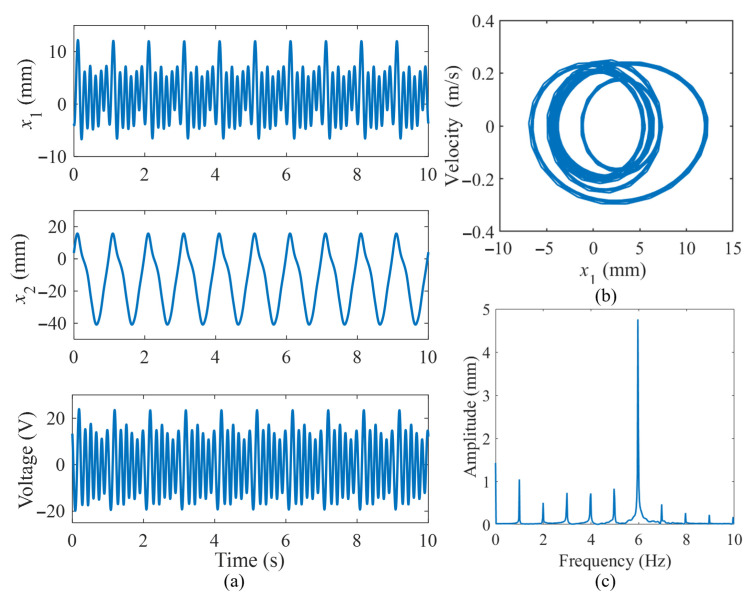
Dynamic response of the system under external excitation: (**a**) the output of displacement and voltage responses; (**b**) phase portraits of displacement *x*_1_; (**c**) frequency spectrum.

**Figure 8 micromachines-16-00942-f008:**
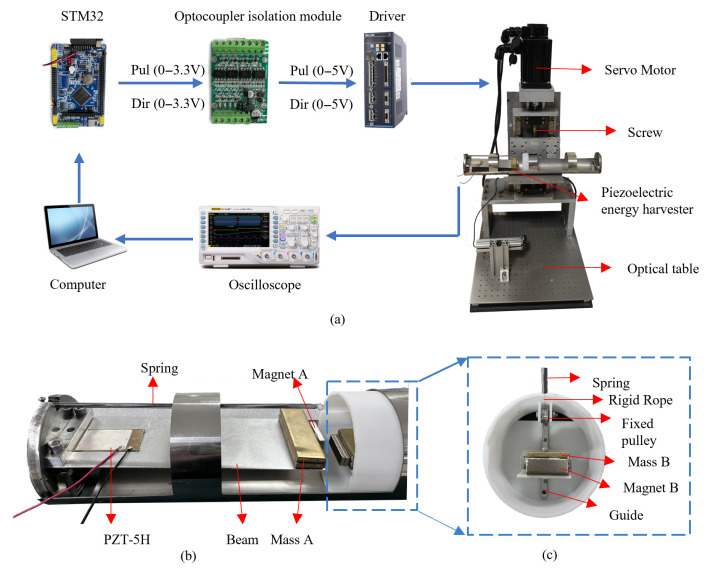
Experimental setup: (**a**) schematic diagram; (**b**) the prototype of harvester; (**c**) spring mass system.

**Figure 9 micromachines-16-00942-f009:**
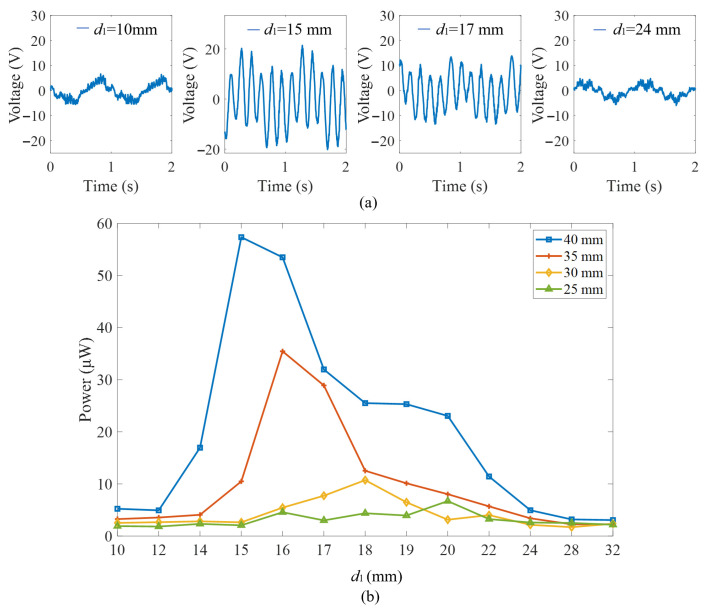
Experimental output varying magnetic distances and excitation amplitudes: (**a**) output voltage; (**b**) output power.

**Figure 10 micromachines-16-00942-f010:**
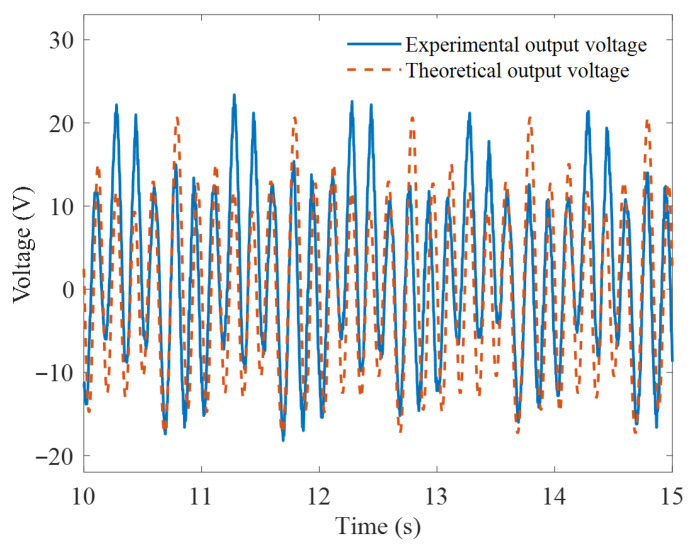
Comparison between the experimental and theoretical output voltages.

**Figure 11 micromachines-16-00942-f011:**
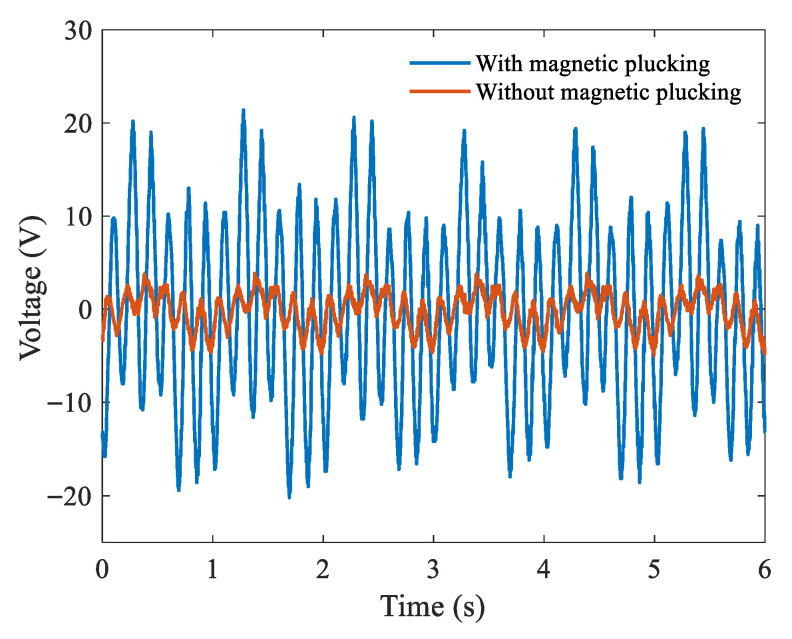
The voltage outputs with and without magnetic plucking.

**Figure 12 micromachines-16-00942-f012:**
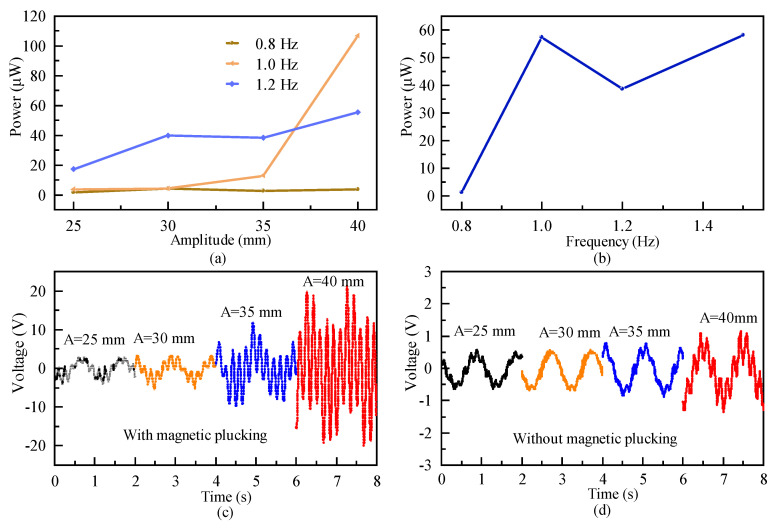
(**a**) Output power at different excitation amplitudes and frequencies; (**b**) output power at A = 40 mm under different frequencies; (**c**,**d**) comparison of output voltage with and without plucking at different amplitudes.

**Figure 13 micromachines-16-00942-f013:**
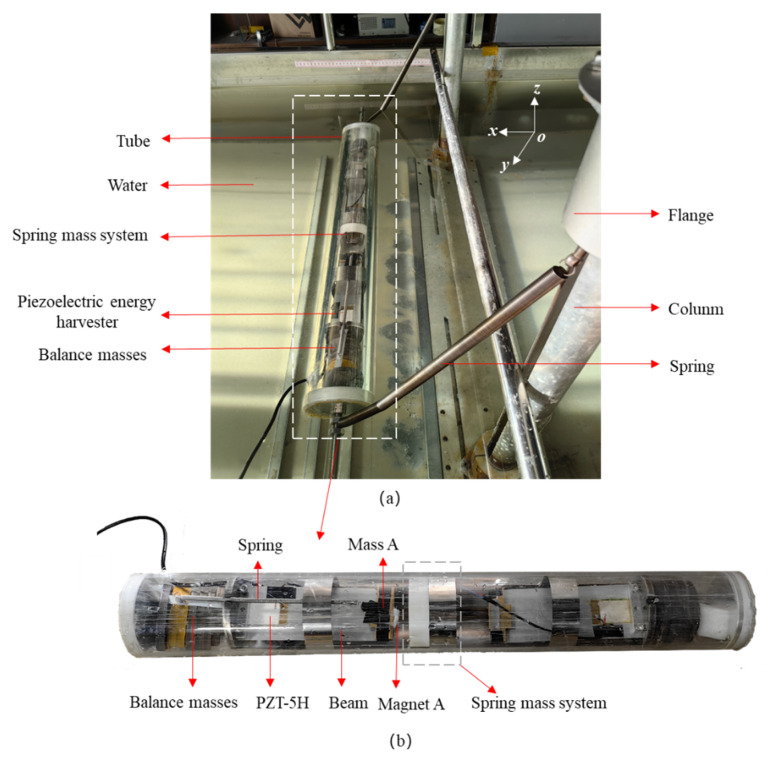
Underwater experimental setup: (**a**) piezoelectric energy harvester in the water channel; (**b**) the prototype of harvester embedded in the tube.

**Figure 14 micromachines-16-00942-f014:**
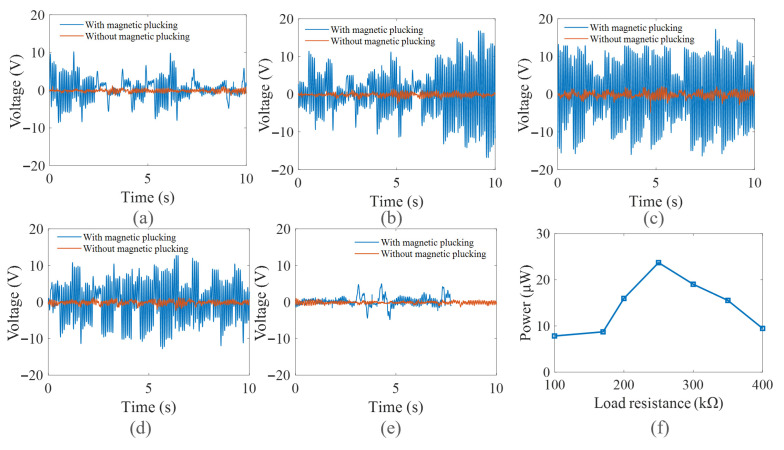
Output voltage and output power at different flow rates: (**a**) 0.313 m/s; (**b**) 0.351 m/s; (**c**) 0. 371 m/s; (**d**) 0.404 m/s; (**e**) 0.431 m/s; (**f**) output power versus external load resistance at a flow rates of 0.371 m/s.

**Table 1 micromachines-16-00942-t001:** Parameters.

Parameters	Symbol	Value	Parameters	Symbol	Value
Mass(copper)			Magnet (NdFeB)		
Length	*l_B_*	30 mm	Length	*l_n_*	8 mm
Width	*w_B_*	30 mm	Width	*w_n_*	30 mm
Height	*h_B_*	20 mm	Thickness	*h_n_*	10 mm
Cantilever (aluminum)			PZT-5H		
Length	*L*	200 mm	Length	*l_p_*	60 mm
Width	*w_b_*	70 mm	Width	*w_p_*	30 mm
Thickness	*h_b_*	1 mm	Thickness	*h_p_*	0.4 mm
Modulus of elasticity	E_b_	72 GPa	Density	*ρ* _p_	7.5 × 10^3^ kg/m^3^
Density	*ρ_b_*	2.7 × 10^3^ kg/m^3^	Modulus of elasticity	E_p_	56 GPa
Other parameters			piezoelectric constant	*d* _31_	−275 pC/N
Remanent flux density	Br	1.21 T	Other parameters		
Stiffness of spring	*k* _eq2_	20 N/m	correction factor	*μ* _1_	1.0344
equivalent damping	*c* _eq1_	0.1802	Vacuum permeability	*μ* _0_	4π × 10^−7^ N/A2
Circuit capacitance	$C_{1}^{s}$	65 × 10^−9^ F			

## Data Availability

The original contributions presented in this study are included in the article. Further inquiries can be directed to the corresponding author.

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
