# Peer review of "Study of an Ultra-Low-Frequency Inertial Vibration Energy Harvester with a Frequency Up-Conversion Approach"

_micromachines, 2025, doi:10.3390/mi16080942_

Round 1
Reviewer 1 Report
Comments and Suggestions for Authors
In this manuscript titled "Study of an ultra-low frequency inertial vibration energy harvester with a frequency up-conversion approach," the authors propose an ultra-low-frequency inertial piezoelectric vibration energy harvester based on a spring–mass system with magnetic plucking for frequency up-conversion. Major revisions are required before this manuscript can be considered for publication in Micromachines.
- In Fig. 7(c), "frequnecy" should be corrected to "frequency." The manuscript should be carefully checked for typographical and grammatical errors.
- The authors conducted a theoretical analysis of the magnetic force between magnets A and B; however, the actual magnetic force was not measured. It is recommended to experimentally measure the magnetic force between the two magnets as a function of distance and angle to validate the theoretical model.
- The novelty is clear, but the authors should better emphasize the differences and advantages compared with existing magnetic frequency up-conversion (FUC) methods.
- Figures 3–5 are informative, but the discussion on the energy barrier heights remains qualitative. A more quantitative analysis is recommended.
- A performance comparison with other recent ultra-low-frequency energy harvesters should be added to strengthen the claims of superiority.
- For practical applications, more discussion on the long-term durability and sealing reliability of the harvester in underwater conditions is needed.
- The presented results are based on short-term measurements. The authors should include data or discussions on the system’s output performance under long-term operation.
- In Fig. 12(c), the voltage axis scales are inconsistent. The authors should consider plotting the output voltages at different amplitudes in a single graph, using different colors to distinguish them. This would make the comparison clearer.
Author Response
Comment 1: In Fig. 7(c), "frequnecy" should be corrected to "frequency." The manuscript should be carefully checked for typographical and grammatical errors.
Response 1: Thank you for pointing out this error. We have modified "frequnecy" in Fig. 7(c) and checked the typographical and grammatical errors in the whole manuscript.
Comment 2: The authors conducted a theoretical analysis of the magnetic force between magnets A and B; however, the actual magnetic force was not measured. It is recommended to experimentally measure the magnetic force between the two magnets as a function of distance and angle to validate the theoretical model.
Response 2: Thank you for your insightful comment. We agree that measuring the magnetic force between magnets A and B is important for verifying the analysis. However, due to the dynamic characteristics of the interaction and limitations of our current experimental setup, direct measurement of the magnetic force is not accomplished at this stage.
Comment 3: The novelty is clear, but the authors should better emphasize the differences and advantages compared with existing magnetic frequency up-conversion (FUC) methods.
Response 3: Thank you for your valuable comment. We appreciate your recognition of the novelty of our work. Unlike many existing frequency up-conversion designs, our approach takes environmental constraints and spatial limitations into account in addition. The proposed harvester was embedded in a cylindrical tube and incorporated a pulley mechanism.
Comment 4: Figures 3–5 are informative, but the discussion on the energy barrier heights remains qualitative. A more quantitative analysis is recommended.
Response 4: Thank you for your valuable comment. We agree that a more quantitative analysis of the energy barrier heights would enhance the depth of the discussion. In the revised manuscript, we have supplemented the original qualitative analysis with a quantitative evaluation of the potential energy barrier heights. Detailly, in Figure 3, when x2=0, the potential energy barrier height is 0.06 J. At other positions, the potential energy surface becomes asymmetric and exhibits dynamic characteristics. In Figure 4, the potential energy barrier height for the beam reaches a maximum of 0.173 J and a minimum of 0.06 J, showing significant variation. In contrast, the spring system displays a different energy profile. For Figure 5, when the magnetic distance d1=7 mm, the potential energy barrier height is 0.09 J. It gradually decreases with the increasing magnetic distance until the system transitions to the monostable state. We have added this analysis in Section 3.1, and highlighted the corresponding changes in the revised manuscript.
Comment 5: A performance comparison with other recent ultra-low-frequency energy harvesters should be added to strengthen the claims of superiority.
Response 5: Compared to other recent ultra-low-frequency energy harvester designs, in which the beams were often oriented parallel to the direction of gravity while the influence of the gravity can be neglected, the beams studied in this manuscript were oriented perpendicular to the direction of gravity and the influence of the gravity can not be neglected. As a result, a direct performance comparison with those designs is not appropriate. We have incorporated a simple comparison of our design’s performance with alternative approaches without magnetic plucking in the manuscript.
Comment 6: For practical applications, more discussion on the long-term durability and sealing reliability of the harvester in underwater conditions is needed.
Response 6: Thank you for your valuable comment. We will consider and study the long-term durability and sealing reliability of the harvester in underwater conditions in the future. At this stage, our focus is to find out the feasibility of the designed harvester.
Comment 7: The presented results are based on short-term measurements. The authors should include data or discussions on the system’s output performance under long-term operation.
Response 7: We agreed that the long-term measurements of the underwater experiments is much better. However, in the present experimental setup, the water channel cannot operate continuously for long time. As a result, we conducted tests lasting several minutes and measured the output. We will address the challenges in the future work.
Comment 8: In Fig. 12(c), the voltage axis scales are inconsistent. The authors should consider plotting the output voltages at different amplitudes in a single graph, using different colors to distinguish them. This would make the comparison clearer.
Response 8: We apologize for the poor quality of the figure. Figure 12c has been replaced in the revised manuscript according to your suggestion. Due to the large difference in voltage amplitude range between them, using the same coordinate axes would make it difficult to distinguish their variations when plotted together.

Figure 12. (a) output power at different excitation amplitudes and frequencies; (b) output power at A=40 mm under different frequencies; (c) and (d) are the comparison of output voltage with and without plucking at different amplitudes.
Reviewer 2 Report
Comments and Suggestions for Authors
The research motivation of this manuscript is clear, and it presents a noteworthy technological advancement in harvesting energy from low-speed water flow. The structural design is innovative, particularly in how it addresses spatial constraints and magnetic coupling through well-defined modeling. However, given the complexity of the magnetic interaction formulas, it is recommended that the authors supplement their explanation with intuitive schematic illustrations showing step-by-step motion.
It is also suggested that the Introduction section include a more thorough comparison of relevant literature, especially regarding differences and advantages over existing underwater piezoelectric harvesters. In the experimental section, further integration and comparison with the earlier theoretical analysis would strengthen the study. Additionally, discussing the effects of flow rate fluctuations and water pressure under varying conditions would enhance the understanding of the device's performance.
Author Response
Comment 1: It is also suggested that the Introduction section include a more thorough comparison of relevant literature, especially regarding differences and advantages over existing underwater piezoelectric harvesters. In the experimental section, further integration and comparison with the earlier theoretical analysis would strengthen the study. Additionally, discussing the effects of flow rate fluctuations and water pressure under varying conditions would enhance the understanding of the device's performance.
Response 1: Thank you for your insightful suggestion! We have summarized recent literature on existing underwater piezoelectric harvesters in Lines 52 to 60. The problem you proposed is critical in practical engineering applications. So far, we are still at the stage of the feasibility study and the prototype validation. We will address the challenges encountered such as aquatic weeds, flow rate fluctuations, water pressure, marine organisms, and biofouling in the future work.
Reviewer 3 Report
Comments and Suggestions for Authors
Review on: Study of an ultra-low frequency inertial vibration energy harvester with a frequency up-conversion approach (micromachines – 3789581)
In the manuscript a vibration energy harvester with a frequency up-conversion approach is proposed for harvesting energy from fluid flow. The manuscript is well structures and includes 5 main chapters. The introduction provides the reader with the necessary information on the topic. In the modeling and simulation section a system model is derived in sufficient detail. Theoretical analysis was performed on all important aspects. Authors also described the experimental setup appropriately. Results are well presented and discussed in sufficient detail.
Following a list of points for further improvement of the manuscript:
Section 2.3: To the reader it is not clear what the difference is between d1 and d. Also, the points A and B are described as the magnetic dipole of magnet A and B. At the same time point A and B are the intersections of the cantilevers’ horizontal extension line with the vertical line of magnet A and B. Thus, in Fig. 2 point B should be directly above magnet B and thus d should be always equal to d1, but is not. Please clarify and, if necessary, use different letters for the dipoles and the intersection points.
Figure 2: x3 should be x2. In the text there is no description for x3.
Line 344: the sentence starting with “Analyze the influence of …” seems to be a request for something. Please check this sentence.
At several positions in the text the output power is given. What load resistance was used to measure the output power?
Figure 12a: what load resistance was used?
Figure 14: please indicate in the figure or figure caption what voltage curves are displayed. There is an orange and a blue curve for presumably with and without plucking.
The excitation in the water experiment was realized with a bluff body. This bluff body cannot be seen in figure 13. What kind of bluff body was used?
Magnetic plucking has been used for a long time to create bistable energy harvesting systems and is therefore not a very new aspect. How does the system proposed by the authors compare to other low frequency energy harvesting systems with bistable characteristic?
Author Response
Comment 1: Section 2.3: To the reader it is not clear what the difference is between d1 and d. Also, the points A and B are described as the magnetic dipole of magnet A and B. At the same time point A and B are the intersections of the cantilevers’ horizontal extension line with the vertical line of magnet A and B. Thus, in Fig. 2, point B should be directly above magnet B and thus d should be always equal to d1, but is not. Please clarify and, if necessary, use different letters for the dipoles and the intersection points.
Response 1: Thank you for pointing out these inappropriate naming notations. We use point A’ and B’ to express the intersections of the cantilevers’ horizontal extension line with the vertical line of magnet A and B. Figure 2 has been modified as you suggested and updated in the revised manuscript.
Comment 2: Figure 2: x3 should be x2. In the text there is no description for x3. Line 344: the sentence starting with “Analyze the influence of …” seems to be a request for something. Please check this sentence.
Response 2: Thank you for pointing out these errors. In Figure 2, x3 has been corrected to x2. The sentence in Line 344 has been revised as follows: Furthermore, the output power of energy harvesters with different magnetic distances should be analyzed under an excitation amplitude of 40 mm. We have highlighted this sentence in the revise manuscript.
Comment 3: At several positions in the text the output power is given. What load resistance was used to measure the output power? Figure 12a: what load resistance was used? Figure 14: please indicate in the figure or figure caption what voltage curves are displayed. There is an orange and a blue curve for presumably with and without plucking.
Response 3: Thank you for your insight comments! As you mentioned, we have calculated the output power in several positions. During the underwater test, a series of load resistances were used to determine the optimal resistance and the maximum underwater power output was measured. In other tests, the equivalent load resistance, estimated as Ropt=1/(ωnCp), was used to calculate the output power [1]. As suggested, we have added a line legend in Figure 14 to clarify the results with and without magnetic plucking.
[1] Liao W.; Wen Y.; Kan, J.; Huang X.; Wang, S.; Li Z.; Zhang Z. A joint-nested 1structure piezoelectric energy harvester for high-performance wind-induced vibration energy harvesting[J], International Journal of Mechanical Sciences, 2022, 227:107443.
Comment 4: The excitation in the water experiment was realized with a bluff body. This bluff body cannot be seen in figure 13. What kind of bluff body was used?
Response 4: The bluff body employed in this study is a cylindrical tube, as shown in Figure 13b. In Line 131, the tube is introduced in the sentence “To harness the flow induced motion energy, a tube cylinder is used. The sealed tube forms a waterproof structure to protect the inside energy harvesters.”
Comment 5: Magnetic plucking has been used for a long time to create bistable energy harvesting systems and is therefore not a very new aspect. How does the system proposed by the authors compare to other low frequency energy harvesting systems with bistable characteristic?
Response 5: Thank you for the comments and suggestion! Other low-frequency energy harvesters may not be suitable for being embedded within a bluff body or integrated with the flow induced vibrations in low-velocity water flow. In contrast, our proposed inertial harvester operates effectively at relatively low flow speeds and features a good waterproof structure. We have not compared it with other designs, because we have not found a similar design with an inertial vibration energy harvester in low-velocity water flow.

Reviewer 4 Report
Comments and Suggestions for Authors
The paper presents an interesting model for energy recovery. However, several improvements should be introduced and clarified:
-
The Introduction should focus on the rationale for selecting the piezoelectric harvester before discussing the electromagnetic option. For low-frequency applications (below 2 Hz), electromagnetic harvesters are generally considered more effective. Please include references to studies that support the effectiveness of piezoelectric harvesters below 2 Hz.
-
The correction factor applied to the forcing function is not clearly explained. Please elaborate on its derivation and physical meaning.
-
In Equation (1), clarify the sign of the electromechanical coupling term. Under what conditions is it positive, and when is it negative?
-
How does the orientation of the magnets affect the results? Please compare the NS–NS and NS–SN configurations and specify which is more advantageous in this context.
-
Please provide a detailed explanation of Equation (13), including its relevance and how it was derived.
-
Consider adding a numerical resonance curve at the beginning of Section 3.2 to demonstrate that the energy harvester is effective at low frequencies.
Comments on the Quality of English Language
The English in certain sections requires improvement.
Author Response
Comment 1: The Introduction should focus on the rationale for selecting the piezoelectric harvester before discussing the electromagnetic option. For low-frequency applications (below 2 Hz), electromagnetic harvesters are generally considered more effective. Please include references to studies that support the effectiveness of piezoelectric harvesters below 2 Hz.
Author response: Thank you for your valuable comment. As you noted, while electromagnetic harvesters are often regarded as more effective at low frequencies due to their ability to generate higher output voltages, piezoelectric harvesters offer some advantages, particularly in compact structures where integration, simplicity, and durability are essential. In this study, the bluff body vibrates along the direction of gravity. Considering the ultra-low-frequency inertial vibration energy harvesting, electromagnetic harvesters would need to overcome gravitational forces, which may make the design more challenging.
Comment 2: The correction factor applied to the forcing function is not clearly explained. Please elaborate on its derivation and physical meaning.
Author response: Considering the deviation of lumped parameter from the distributed parameter model, the correction factor μ1 is derived for compensating the calculation results. From the Erturk’s theory [2], it can be expressed as follows:

where Mti is the tip mass of the harvester, mi is the mass of the cantilever beam.
[2] Erturk A.; Inman D J. On Mechanical Modeling of Cantilevered Piezoelectric Vibration Energy Harvesters[J]. Journal of Intelligent Material Systems and Structures, 2008, 19(11): 1311-1325.
Comment 3: In Equation (1), clarify the sign of the electromechanical coupling term. Under what conditions is it positive, and when is it negative?
Author response: In Equation (1), the electromechanical coupling term is negative. The sign of this term depends on the relationship between the poling direction of the piezoelectric material and the direction of the applied mechanical stress or strain. If they are opposite, the coupling term is negative; if they are in the same direction, it is positive.
Comment 3: How does the orientation of the magnets affect the results? Please compare the NS-NS and NS-SN configurations and specify which is more advantageous in this context.
Author response: Thank you for your insightful question regarding the effect of magnet orientation on the system's performance. Based on the literature on ultra-low-frequency energy harvesting by Gao et al. [3], the magnetic repulsive force configuration was selected for the bistable energy harvester. This repulsive (NS-SN) arrangement is particularly suitable for inducing bi-stability under low-frequency excitation. Therefore, we used the repulsive configuration.
[3]. Gao Y.J.; Leng Y.G.; Fan S.B. and Lai Z.H. Performance of bistable piezoelectric cantilever vibration energy harvesters with an elastic support external magnet. Smart Mater. Struct. 2014, 23: 095003.
Comment 4: Please provide a detailed explanation of Equation (13), including its relevance and how it was derived.
Author response: Thank you for your comment! The vector product is calculated by using the projection method, and the projection on Figure 2 demonstrates that
, By substituting these relations into Equation (13), the derivation can be significantly simplified.

Figure 2. Geometric analytical model of the two magnetic.
Comment 5: Consider adding a numerical resonance curve at the beginning of Section 3.2 to demonstrate that the energy harvester is effective at low frequencies.
Author response: Thank you for your suggestion! We agree that resonance curve would enhance the understanding of the harvester’s frequency response. In our design, effective low-frequency energy harvesting is achieved through magnetic excitation, enabling frequency up-conversion with a ratio of approximately 1:6. We believe that an experimental resonance curve is more persuasive than a numerical one. Therefore, we conducted three free-decay experiments to obtain the experimental resonance curves. The results presented below in Figure 3 indicate a resonant frequency of 6 Hz.

Figure 3. The resonance curve of energy harvester.

Round 2
Reviewer 2 Report
Comments and Suggestions for Authors
The revision is good. The journal can accept the manuscript.
Reviewer 4 Report
Comments and Suggestions for Authors
The revised paper look much better. All my questions were explained and introduced in the manuscript. Therefore, I can recommend it for publication.